# Patterns of metastases progression- The linear parallel ratio

Ofer N. Gofrit[1]*, Ben Gofrit[2], Yuval Roditi[2], Aron Popovtzer[3], Steve Frank[3], Jacob Sosna[4], S. Nahum Goldberg[4]

1 Department of Urology, Hadassah Medical Center, Faculty of Medicine, Hebrew University of Jerusalem, Jerusalem, Israel, 2 School of Engineering and Computer Science, Hebrew University of Jerusalem, Jerusalem, Israel, 3 Department of Oncology, Hadassah Medical Center, Faculty of Medicine, Hebrew University of Jerusalem, Jerusalem, Israel, 4 Department of Radiology, Hadassah Medical Center, Faculty of Medicine, Hebrew University of Jerusalem, Jerusalem, Israel

* ogofrit@gmail.com

## Abstract

**Data Availability Statement:** The complete data set is available in S1 Dataset.

**Funding:** The authors received no specific funding for this work.

### Background

Linear and parallel are the two leading models of metastatic progression. In this study we propose a simple way to differentiate between them. While the linear model predicts accumulation of genetic and epigenetic alterations within the primary tumor by founder cells before spreading as waves of metastases, the parallel model suggests preclinical distribution of less advanced disseminated tumor cells with independent selection and expansion at the ectopic sites. Due to identical clonal origin and time of dispatching, linear metastases are expected to have comparable diameters in any specific organ while parallel metastases are expected to appear in variable sizes.

### Methods and findings

Retrospective revision of chest CT of oncological patients with lung metastases was performed. Metastasis number and largest diameters were recorded. The sum number of metastases with a similar diameter (c) and those without (i) was counted and the linear/parallel ratio (LPR) was calculated for each patient using the formula $(\sum c - \sum i)/(\sum c + \sum i)$. A LPR ratio of 1 implies pure linear progression pattern and -1 pure parallel. 12,887 metastases were measured in 503 patients with nine malignancy types. The median LPR of the entire group was 0.71 (IQR 0.14–0.93). In carcinomas of the pancreas, prostate, and thyroid the median LPR was 1. Median LPRs were 0.91, 0.65, 0.60, 0.58, 0.50 and 0.43 in renal cell carcinomas, melanomas, colorectal, breast, bladder, and sarcomas, respectively.

### Conclusions

Metastatic spread of thyroid, pancreas, and prostate tumors is almost exclusively by a linear route. The spread of kidney, melanoma, colorectal, breast, bladder and sarcoma is both linear and parallel with increasing dominance of the parallel route in this order. These findings

**Competing interests:** None of the authors have any direct or indirect financial incentive associated with publishing the article.

can explain and predict the clinical and genomic features of these tumors and can potentially be used for evaluation of metastatic origin in the individual patient.

## Introduction

Cancer is the second leading cause of death in the Western World. More than 90% of cancer related deaths are attributed to metastases [1]. Understanding this process is critical for improving patient care. To become a metastasis, a cancer cell must gain multiple capabilities including progressive growth, vascularization, invasion, detachment, embolization, survival in the circulation, arrest, extravasation, evasion of host defense and progressive growth at the landing site [2]. While primary tumors are heterogenous both genotypically and phenotypically and contain diverse subpopulations, metastases are often less so due to the multiple bottlenecks that they subsequently cross [3–6].

The two leading models of metastatic progression are the linear and the parallel [7–9]. The linear model predicts accumulation of genetic and epigenetic alterations and selection for competitive fitness within the primary tumor. Once all necessary abilities are gained by a founder cell (or cluster of cells), the invasion-metastasis cascade is initiated, and a wave of metastases ensues [10]. Further waves can arise later when additional founder cells gain the necessary capabilities. The parallel model suggests preclinical distribution of less advanced disseminated tumor cells (DTCs). Selection and expansion then occur independently at the ectopic sites and in parallel to the primary tumor.

Distinction between the linear and parallel models is important both scientifically and clinically. The linear model suggests a higher degree of similarity between the primary tumor and its metastases and therefore, genetic analysis of the primary tumor is a reliable surrogate of the metastases genome and response to therapy. The parallel model, on the other hand, suggests a higher degree of genomic diversity between the metastases and the primary tumor and among the metastases themselves, potentially requiring more aggressive therapy [11]. The linear model suggests that early eradication of the primary tumor could prevent metastatic dissemination, whereas in the parallel model systemic therapy should be the first priority.

Lineage tracing of metastases is based on phylogenetic comparisons of primary tumors and their metastases using technologies such as somatic copy-number alterations, single nucleotide variants, microsatellite analysis, epigenetic studies, and whole-genome analysis. The information they yield is limited by many factors including tissue availability, non-synchronous tissue collection, mutational burden of the tumor founder cell, potential convergent evolution, potential loss of precursor cells in the primary tumor, potential of not assessing the area harboring the pre-metastatic clone in the primary tumor, self-seeding, and selection pressures induced by therapy. The literature concerning these studies is rich and exciting, yet it is often complex and confounding [8–12].

Our hypothesis is that differentiation between the linear and parallel routes of metastatic spread can be done by measurements of metastases diameters on the clinical CT (Fig 1). The hypothesis is based on the following assumptions:

1. If the linear route dominates, then once a clone has gained all the necessary properties, it will spread as a wave of metastases into host organs. Due to identical clonal origin, similar conditions at the host organ and similar kinetics, these metastases are expected to grow at a similar rate and have comparable diameters for any specific organ. If further clones reach maturity, then ensuing waves are expected. Thus, metastases could present with different sizes, but all can be fitted into a few, discrete clusters of sizes.

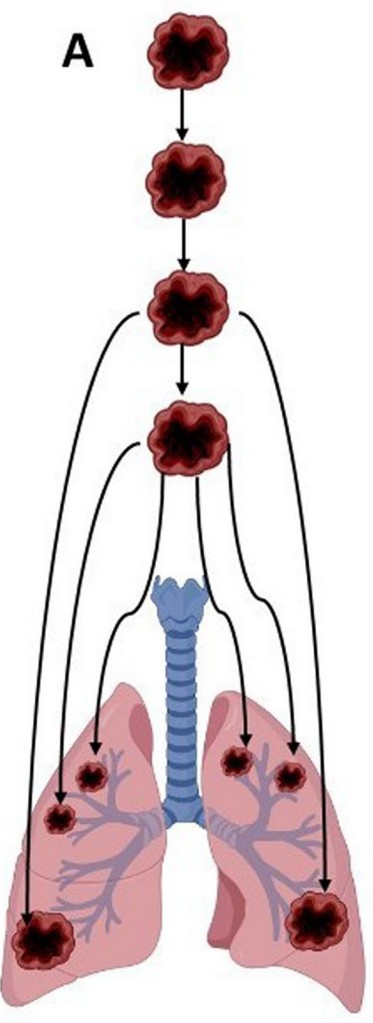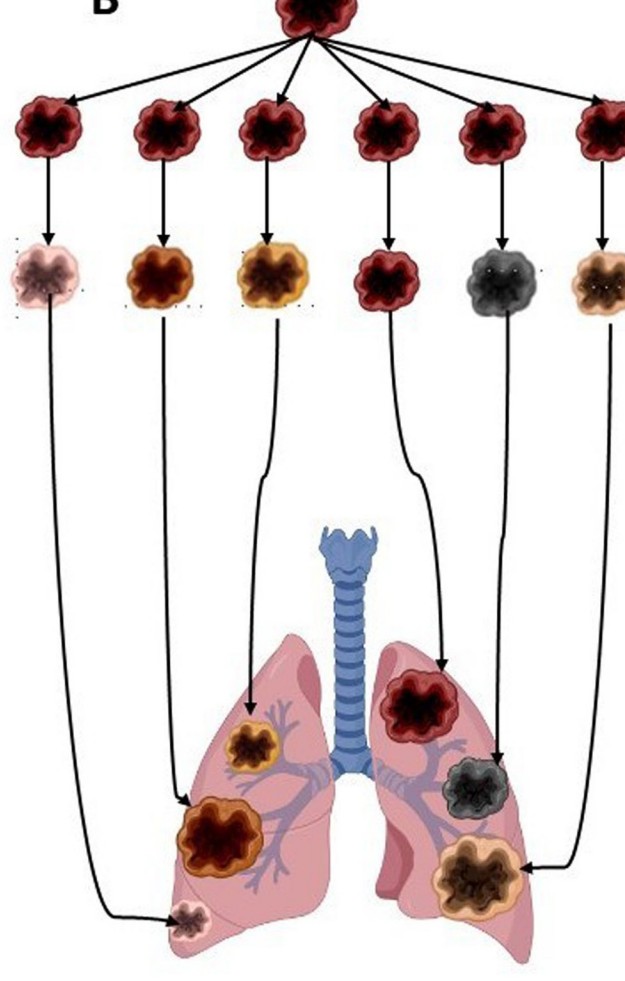

**Fig 1. The linear and the parallel models of metastases progression.**

2. If the parallel route prevails, then there is an early spread of DTCs that gain metastatic capabilities independently. Differences in extravasation time and growth rates are expected to produce metastases of variable sizes that cannot be fitted into one or few clusters, but rather present as a continuous range of diameters.

   a. The linear model. Metastatic cells mature in the primary tumor and spread as a single or as several waves. Due to identical clonal origin, similar conditions in the host organ and similar date of commission, these metastases are expected to grow at a similar pace and be of similar diameter. Their linear/parallel ratio (LPR) should be 1.

   b. The parallel model suggests preclinical distribution of less advanced disseminated tumour cells (DTCs). Selection and expansion then occur independently at the ectopic sites and in parallel to the primary tumour. Differences in implantation date and growth rate are expected to produce metastases of different sizes that cannot be fitted into a

model of few clusters. Hence, their linear/parallel ratio (LPR) should be -1 (The figure was created under license using the BioRender software).

Lung metastases are usually discrete and oval and are readily measurable from chest CT images against the low-density lung parenchyma. Therefore, we focused on lung metastases to test our hypothesis. To this end, the linear/parallel ratio (LPR) was developed as an indicator of the metastatic spread pattern. LPR ratio of 1 implies pure linear progression, whereas -1 implies a pure parallel one. Nine types of primary tumors that commonly metastasize to the lungs were selected, including carcinomas of the kidney, bladder, prostate, breast, thyroid and pancreas, melanoma, and sarcoma.

## Methods

### Patients

The database of a tertiary hospital center with a comprehensive cancer program was searched for patients diagnosed with lung metastases (ICD-9-CM code 197.0) over the years 2010–2020. The records of these patients were reviewed for the following parameters: age, gender, type of malignancy, date of cancer diagnosis, date of metastases diagnosis, and fate of the patients. The study was approved by the IRB of the Hadassah Medical Organization (# 0650-21-HMO) All data were fully anonymized before analysis and the IRB waived the requirement for informed consent.

Patients included in the study had:

1. Diagnosis of one of the following primary tumors: colorectal, breast, kidney, thyroid, bladder, prostate, thyroid, sarcoma, or melanoma.

2. 2. Two or more lung metastases defined as discrete, round or ellipsoid lesions inside the lung parenchyma, measuring ≥3 mm in their largest diameter.

Exclusion criteria:

1. Patients with history of primary lung cancer.

2. Patients with more than one primary cancer.

The axial chest CT scans of 503 patients, obtained using voltage setting of 120KVp, amperage of 260–330 mA and slice thickness reconstruction of 3 mm or less, were reviewed. The patient's last CT was selected for interpretation. Nonetheless, earlier surveys were examined in 26 patients (5.2%), whenever the last CT was uninterpretable due to confounding variables such as large pleural effusion, lung collapse, coalescence of multiple metastases that precluded their individual measurement or whenever curative therapy was successful (e.g., radioactive Iodine in metastatic carcinomas of the thyroid). For each study, metastasis largest diameter was determined manually, by placing electronic calipers at the margins of the tumor. The number and the largest diameter of all the metastases were recorded.

### Statistics

For each patient, lung metastases that appeared in clusters of similar diameters (i.e., with a diameter deviation $\leq 1$ mm) were counted and their sum number was marked as $\Sigma c$. The sum number of metastases that cannot be classified (isolated) into a similar size category was marked as $\Sigma i$. The linear/parallel ratio (LPR) was calculated for each patient using the formula

$(\Sigma c-\Sigma i)/(\Sigma c+\Sigma i)$ with the aid of a Python written computer code (S1 File). LPR ratio of 1 suggests pure linear progression and -1 pure parallel.

## Results

The study is based on analysis of the chest CT scans of 503 patients demonstrating 12,887 lung metastases. These included 27 patients with thyroid cancer, 26 with prostate cancer, 30 with pancreatic cancer, 45 with kidney cancer, 48 with melanoma, 149 with colorectal cancer, 72 with breast cancer, 35 with bladder cancer, and 71 with sarcomas. Their basic characteristics and outcomes are presented in S1 and S2 Tables, respectively. The complete data set is provided in S1 Dataset.

About half of the patients presented with lung metastases upon initial diagnosis and half developed them during follow-up. Eighty-one percent of the patients died during a median follow-up of 34 months. In 98.7% of them, death was disease specific. Patients with thyroid cancer had the highest number of metastases (average of 42.6 per patient, SD 43.4), and patients with breast cancer the lowest (average of 13.3 per patient, SD 13.1). The largest metastases were noted in sarcoma patients (average diameter of 15.3 mm, SD 8.9) and the smallest in prostate cancer patients (average diameter of 7.6 mm, SD 2.6). Metastases of different primary tumors showed different geomorphological and size distribution patterns (Figs 2 and 3).

Table 1 and Fig 4 show parameters of the metastases in the various tumor types including the linear/parallel ratio (LPR). The median LPR of the entire group was 0.71 (IQR 0.14–0.93).

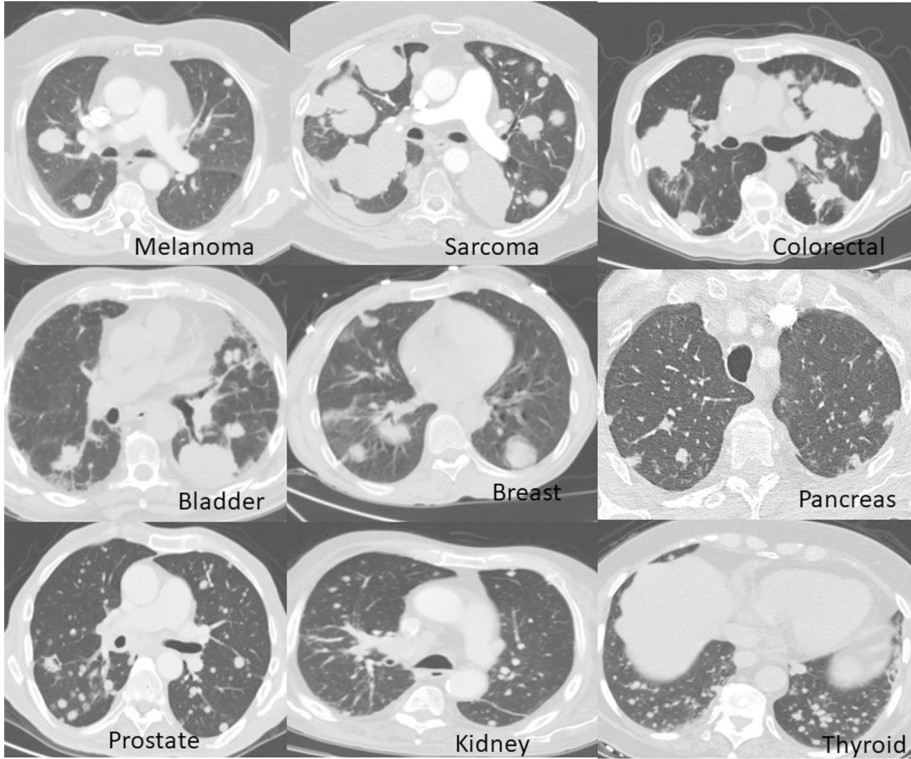

**Fig 2. Representative chest CT scans of patients with lung metastases.** Patients with melanomas, sarcomas, colorectal, bladder and breast cancers typically show variable diameters of metastases, while patients with pancreas, prostate, kidney, and thyroid cancers typically show metastases of similar diameters.

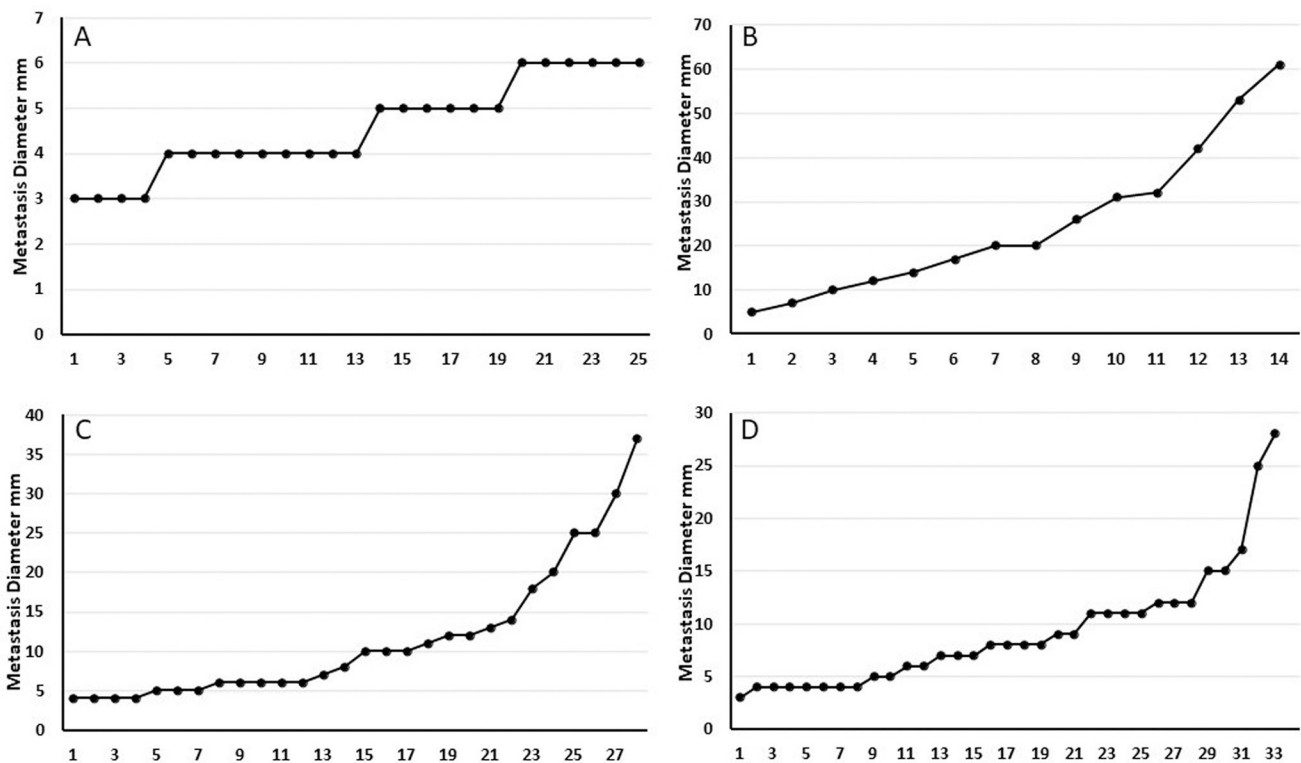

**Fig 3. Examples of metastases diameter distribution in various diseases.** A. an 82-year-old man with pancreatic tumor and 25 metastases. Four waves of metastases are seen LPR = 1. B. a 66-year-old man with leiomyosarcoma. Fourteen metastases are depicted with very limited similarity in diameters. LPR = -0.43. C. a 72-year-old man with colorectal carcinoma and 28 metastases. Some of the metastases can be clustered and some cannot, LPR = 0.57. D. a 72-year-old man with melanoma. Thirty-three metastases are depicted. Six to seven waves of metastases can be seen. LPR = 0.82.

Malignancies of the pancreas, prostate, and thyroid showed a remarkably high median LPR of 1. A high median LPR of 0.91 was also noted in kidney cancer. Melanomas, colorectal, breast, and bladder cancers revealed lower LPR ratios (0.65, 0.60, 0.58 and 0.50 respectively), and sarcomas had the lowest LPR of 0.43.

**Table 1. Measurements of the metastases and their linear/parallel ratio (LPR).**

| Primary tumor | Average number of metastases per patient (SD) | Average individual metastases diameter (SD)** | Median linear/parallel ratio (Q1, Q3) |
|---|---|---|---|
| Thyroid | 42.6 (43.4) | 8.2 (4.7) | 1.0 (0.87–1) |
| Pancreas | 23.7 (21.5) | 8.0 (3.6) | 1.0 (0.97–1) |
| Prostate | 23.0 (30.5) | 7.6 (2.6) | 1.0 (0.86–1) |
| Kidney | 23.2 (33.4) | 12.2 (7.9) | 0.91 (0–6,1) |
| Melanoma | 25.5 (31.5) | 11.2 (5.4) | 0.65 (0–0.86) |
| Colorectal | 29.9 (33.5) | 11.4 (5.6) | 0.6 (0.14–0.88) |
| Breast | 13.3 (13.1) | 11.5 (6.9) | 0.58 (0–0.81) |
| Bladder | 23.4 (34.8) | 12.2 (6.2) | 0.5 (-0.42–0.92) |
| Sarcomas | 18.7 (18.7) | 15.3 (8.9) | 0.43 (-0.18–0.75) |
| Overall | 23.5 (29.8) | 11.5 (6.7) | 0.71 (0.14–0.93) |

** in mm

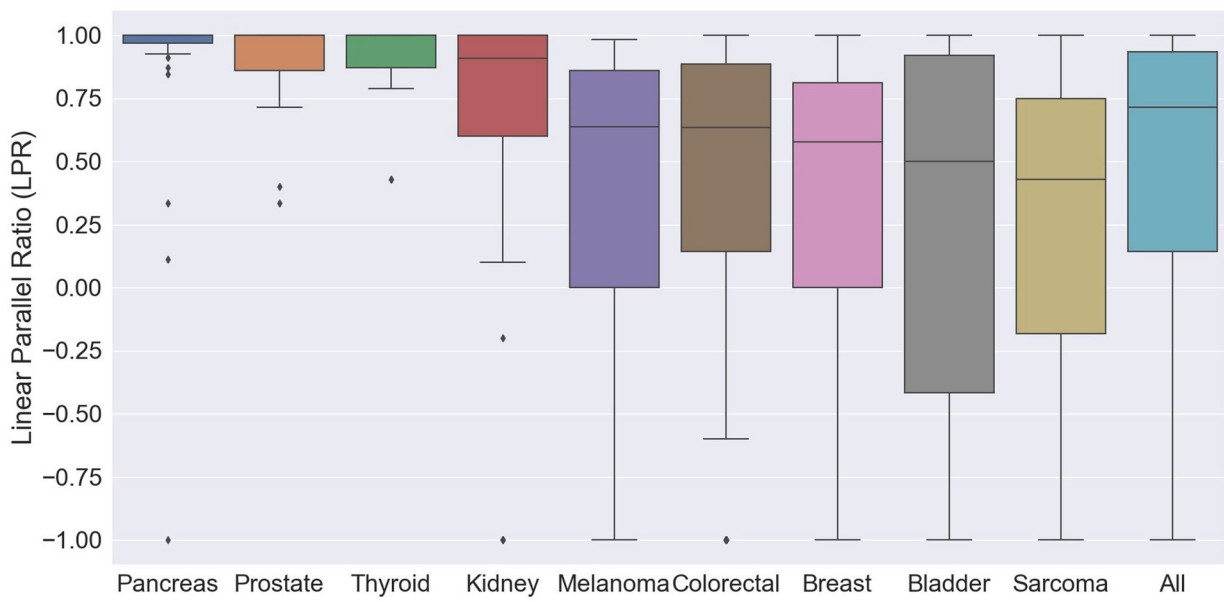

**Fig 4. Whisker plot of linear/parallel ratios of nine malignancies including the outliers of each tumor.**

## Discussion

Metastatic evolution is a highly complex process characterized by potential of early spread of DTCs, repeated waves of metastases, metastasis-to-metastasis spread, periods of dormancy, and different growth rates of polyclonal cells. Attempts to systematise this process has led to two theories, namely the linear and the parallel progression models [7–9, 13].

In our study, we attempted to differentiate the two models using data from 12,887 lung metastases of 503 patients with multiple lung metastases viewed in CT. When the linear model applies, waves of metastases with similar diameters are expected. In the parallel model, metastases should take variable diameters. To assess this, a novel parameter, the LPR, was developed. A LPR of 1 implies pure linear progression and -1 pure parallel. The median LPR of the entire group was 0.71 (IQR 0.14, 0.93), suggesting that among the 9 types of cancers studied, linear spread is more common than parallel. However, striking differences were found between the different tumors (Table 1 and Fig 4). Cancers of the thyroid, pancreas, kidney, and prostate showed a median LPR of 1 or nearly so, with small IQRs and a few outliers. The other tumors studied had lower LPRs, with the lowest being sarcoma and bladder cancer (0.43 and 0.50).

A brief discussion of each type of cancer follows:

### Pancreas

Pancreatic tumors showed an almost pure linear progression pattern with a median LPR of 1. This is supported by genome sequencing of primary pancreatic tumors and their metastatic deposits showing that the clonal population of metastases is represented in the primary tumor [14]. This implies that metastases are a late event in pancreatic cancer evolution, and that early diagnosis and local treatment are potentially curative. Unfortunately, thus far there is no effective mean for early detection of pancreatic cancer.

### Prostate

Here too, a median LPR of 1 suggests that metastatic spread in prostate cancer is almost exclusively by the linear mode. This is in agreement with molecular studies. Primary prostate cancer

is a multifocal disease harboring multiple genomically distinct foci with non-overlapping mutation profiles, suggesting separate and independent evolutionary trajectories [15, 16]. Prostate cancer metastases, on the other hand, show mono or oligoclonality [16–19]. The linear model in prostate cancer is also supported by the clinical findings of the STAMPEDE trial [20]. This study showed that prostate radiotherapy improves overall survival of men with low metastatic burden (presumably, after only one or a few waves of metastases were deployed) by preventing further waves. In patients with high metastatic burden (after multiple waves of metastases), this treatment is less efficient.

### Renal cell Carcinoma (RCC)

The median LPR of kidney cancer was 0.91 pointing toward dominance of the linear mode of progression in this disease, but with existence of at least an occasional component of parallel dissemination mode as well. This is supported by genomic analyses of 575 primary and 335 metastatic lesions in 100 RCC patients by Turajlic et al. These authors showed two types of metastatic progression: rapid progression to multiple tumor sites by primary tumors of mono-clonal structure (linear) and attenuated progression by heterogenous cells (parallel) [21].

The CARMENA clinical trial randomized intermediate and poor-risk metastatic RCC patients to Sunitinib only or to Sunitinib plus cytoreductive nephrectomy. Sunitinib alone was non-inferior to cytoreductive nephrectomy followed by sunitinib [22]. However, significant overall survival benefit (HR 0.34, 95% CI, 0.22–0.54) was gained in patients who underwent secondary nephrectomy, stressing the importance of the primary tumor as a continuous source of further waves of metastases as demonstrated by the linear model.

### Bladder

The median LPR of bladder cancer was 0.50, one of the lowest in the study, suggesting a predominant role of parallel metastasis progression. This is in agreement with studies of genomic classification comparing primary tumors and their metastases. Faltas et al, performed whole exome sequencing and clonality analysis of 16 matched sets of primary and metastatic tumors. They found that metastatic spread was multiple, parallel and occurred early in the natural history of the disease in all cases [23]. The origin of this phenomenon is probably the high mutation burden of this tumor leading to a high interpatient, intratumoral and intertumoral heterogeneity [24]. Additionally, neoadjuvant chemotherapy before radical cystectomy improves survival and is the standard of care in patients with non-metastatic muscle-invasive bladder cancer, further supporting the concept of early, subclinical spread of invasive bladder cancer cells, requiring more than local excision [25].

### Colorectal cancer (CRC)

The current study showed mixed parallel and linear progression patterns of CRC (median LPR 0.60). This is in concordance with genomic analyses of primary colon cancer and its metastases which likewise show conflicting results suggesting that both mechanisms of metastatic spread are active in CRC. Wei et al, using whole-exome sequencing of 28 samples of matched primary and metastatic tumors from four CRC patients showed that all metastases inherited multiple genetically distinct subclones [26]. Similarly, Vermaat et al, using deep sequencing of DNA isolated from primary and subsequent hepatic metastasis of 21 patients, showed substantial genetic differences between the primary tumor and its metastases, with an average gain of 83 potentially function-impairing variants and 70 losses [27].

## Breast

Our study suggests that dissemination of breast cancer is complex and likely includes both linear and parallel progression (LPR 0.58). This is in line with phylogenetic analysis of primary and metastatic breast cancer and with the findings of high level of DTCs with high heterogeneity in the bone marrow [28–30]. It is also supported by the clinical recommendation for neoadjuvant chemotherapy [31]. A parallel model may explain the failure of locoregional treatment to affect overall survival of some women with breast cancer [32]. Thus, distinction of pattern may be of particular importance for the treatment paradigms of this cancer.

## Melanoma

This tumor shows a mixed linear and parallel progression pattern (LPR 0.65). Surgical treatment in early stages of melanoma is associated with excellent prognosis (98.4% 5-year overall survival for stages 0, 1, and 2), suggesting that early parallel spread does not occur in most patients. Yet, in metastatic patients it has been shown that lineage diversification is pervasive, supporting the parallel mode of metastatic spread [33]. Melanoma can also switch between proliferative and invasive states according to environmental conditions further contributing to a disorganized distribution of metastases' diameters in this tumor [34].

## Sarcoma

The LPR of sarcomas was the lowest among the tumors studied (median 0.43), suggesting mixed linear and parallel dissemination. This is in concordance with whole exome and genome analyses of 86 tumor regions in 10 patients with metastatic osteosarcoma. Metastases showed significantly higher mutational burden and genomic instability compared to the primary tumors. Pulmonary spread was linear in 6/10 patients and parallel in 4/10 [35]. It is suggested that neoadjuvant chemotherapy for treating micrometastases be considered before surgical treatment of sarcomas.

## Thyroid

Clinicians have long been aware of the striking number and resemblance of lung metastases in thyroid cancer that are often very small and termed micronodular metastases (Fig 2). This is reflected by a high LPR (median of 1), indicating an almost pure linear progression pattern. Unfortunately, genomic information in this disease is limited. A single study in whole-exome sequencing showed a median concordance of only 38% in somatic mutations between the metastases and the primary tumor but only 3 patients with lung metastases were included in this series [36].

The LPR model is probably an oversimplification of the metastatic process (as any model in biology), yet it can potentially explain, predict and refine phenomena in oncology. A potential example is the oligometastatic state (OS), usually defined as a state of five metastases or less [37, 38]. Cases of oligometastasis can potentially be managed by metastasis-directed therapy without systemic treatment. Yet, patients with parallel spreading tumors, even with less than five detectable metastases are obviously not suitable for local treatment only. Adding a high LPR to the definition of OS will decrease the number of patients with OS but will improve their outcome and will direct patients with non-OS to systemic therapy. As stated previously, formal calculation of the LPR is not mandatory, and the naked eye can easily categorize similar and dissimilar metastases (Fig 2).

## Limitations of the study

Although potentially helpful in the many ways, the LPR model has several limitations worth mentioning:

1. Metastases to the same organ originating in a single clone may grow in different rates due to proximity to nutrients or to an anatomical structure, thus reaching different diameters and falsely decreasing the LPR.

2. In a parallel spreading disease, a later, nevertheless faster growing clone could reach the same size as an earlier but slower growing one, thus reaching the same diameter and falsely increasing the LPR.

3. For the sake of simplicity, metastases diameters were measured from the axial CT slices only; however, this may not necessarily be their largest diameter thus distorting the measurements.

4. The conclusions of this work apply to metastatic implantation in the lung microenvironment. It is not certain that they can be extrapolated to other organs.

5. The analysis is population derived and based on a snapshot and not on sequential images in individual patients that could reveal a different course.

6. Histologic confirmation of CT findings was not performed in most of the cases. However, 81% of the patients in the series died during follow-up and death was disease-specific in 98.7% of them, implying that the lesions seen on chest CT were indeed metastases in most cases.

7. The LPR was calculated using metastases longest axis on the axial views. This is not necessarily the best morphological predictor of the LPR. Other potential parameters such as metastases short axis, multiplication of the axes or perhaps metastases volume may deliver better predictions. We hope that future research will provide this information.

In summary, we showed a high LPR for metastases originating in the thyroid, kidney, pancreas, and prostate. This point toward a linear route of metastatic spread. The LPR of melanoma, colorectal, breast, bladder, and sarcoma points toward mixed linear and parallel routs with increasing relevance of the parallel route in this order. These findings are in line with most of the genomic work performed and with clinical observations. The LPR was calculated here with aid of a computer code, but can easily be observed by the naked eye, which is sensitive to this type of information. Thus, with further confirmation, we envision that the LPR can be applied to individual patients and assist in oncological decision-making. Future directions of studying the LPR model may include verifications of the findings in other databases and other organs (brain and liver metastases), genomic analyses of several metastases removed from the same patients to study if similar size equals similar genomic traits and comparing the growth of individual metastases in sequential CT studies.

## Supporting information

**S1 File. Program code.**
(DOCX)

**S1 Table. Basic parameters of patients with lung metastases.**
(DOCX)

**S2 Table. Outcome of patients with lung metastases.**
(DOCX)

**S1 Dataset. The complete data set.**
(XLS)

## Acknowledgments

We would like to acknowledge Dr. Galith Abourbeh for her scientific support.

## Author Contributions

**Conceptualization:** Ofer N. Gofrit, Ben Gofrit, Yuval Roditi, Aron Popovtzer, Steve Frank, Jacob Sosna, S. Nahum Goldberg.

**Data curation:** Ofer N. Gofrit, Jacob Sosna, S. Nahum Goldberg.

**Formal analysis:** Ofer N. Gofrit, Ben Gofrit, Yuval Roditi, S. Nahum Goldberg.

**Methodology:** Ofer N. Gofrit, Aron Popovtzer, Steve Frank.

**Software:** Ben Gofrit, Yuval Roditi.

**Supervision:** Jacob Sosna.

**Validation:** Ofer N. Gofrit, Aron Popovtzer.

**Writing – original draft:** Ofer N. Gofrit, S. Nahum Goldberg.

**Writing – review & editing:** Ofer N. Gofrit, Aron Popovtzer, Steve Frank, Jacob Sosna, S. Nahum Goldberg.

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
