## [Decision Letter · Decision Letter 0]

20 Jun 2022

PONE-D-22-06443Patterns of Metastases Progression- The Linear Parallel RatioPLOS ONE

Dear Dr. Gofrit,

Thank you for submitting your manuscript to PLOS ONE. After careful consideration, we feel that it has merit but does not fully meet PLOS ONE’s publication criteria as it currently stands. Therefore, we invite you to submit a revised version of the manuscript that addresses the points raised during the review process.

Specifically, I find your manuscript very interesting, as it provides some novel insights into metastasis and the interpretation of CT scans. In accordance with the expert reviewers, there are however, a few concerns that need to be addressed. Rather than repeat those points here, I refer you to the specific remarks (below) for details.

We look forward to receiving your revised manuscript.

Kind regards,

Nikolas K. Haass, MD/PhD

Academic Editor

PLOS ONE

Journal Requirements:

No

No

5. Please include your tables as part of your main manuscript and remove the individual files. Please note that supplementary tables (should remain/ be uploaded) as separate "supporting information" files.

Additional Editor Comments:

I find your manuscript very interesting, as it provides some novel insights into metastasis and the interpretation of CT scans. In accordance with the expert reviewers, there are however, a few concerns that need to be addressed. Rather than repeat those points here, I refer you to the specific remarks (below) for details.

Reviewers' comments:

Reviewer's Responses to Questions

**Comments to the Author**

1. Is the manuscript technically sound, and do the data support the conclusions?

Reviewer #1: Yes

Reviewer #2: Partly

2. Has the statistical analysis been performed appropriately and rigorously? 

Reviewer #1: N/A

Reviewer #2: Yes

3. Have the authors made all data underlying the findings in their manuscript fully available?

Reviewer #1: Yes

Reviewer #2: Yes

4. Is the manuscript presented in an intelligible fashion and written in standard English?

Reviewer #1: Yes

Reviewer #2: Yes

5. Review Comments to the Author

Reviewer #1: The authors present a study examining CT features to trace the lineage of lung metastases. By measuring and comparing the size of lung mets in each patient, they draw conclusions regarding the metastatic progression of each of the 9 diseases included in the study. Some cancer types had a high linear/parallel ratio (LPR) that indicated similarity between the different metastases, while other cancers had low LPR indicating greater variation.

Overall, I find their observations interesting. These results could have clinical relevance as CT scans are routine in the diagnosis and treatment of cancer patients.

I have a few points that I’d like the authors to comment on:

1. For some of the cancers presented here, melanoma and breast cancer for example, there are multiple histological subtypes that are related to different treatments and varied patient outcomes. Perhaps the data is over simplified by combining the subtypes. Can you be more specific about the clinical data presented. Is some of the variation in LPR in theses cancers due to histological subtype?

2. Is there any relationship between LPR and treatment using targeted or immunotherapy?

3. Have you looked for a relationship between LPR and overall patient survival?

4. Does the LPR ratio for each cancer type align with published tumour mutation burden rates for each cancer?

Reviewer #2: Review

This is an interesting study that reports on data obtained from lung CT scans.

Overall, I think it is worth publication in this journal. However, I there are a few issues that I believe the authors should address. Allow me to expand

as follows:

1. It is not clear the authors have a hypothesis even though they state they are performing the study to test their hypothesis. What is the hypothesis?

If there is one, it needs to be stated clearly, and it needs to be stated how these data obtained can address the likelihood of the hypothesis being true.

If this is not possible, then I think the study is best framed as an investigation that provides evidence that supports the hypothesis that

alternative metastatic disease morphologic patterns in lung (and by extension, possibly elsewhere) may reflect alternate pathogenic routes.

And the evidence is the correlation between known genomic data and CT morphologic patterns. Because there is a possible mechanism that may

account for this correlation, it may represent causation, thus a valid hypothesis may exist.

2. The authors provide evidence for such a correlation, but I think this should be the main finding of the investigation, and should be part of the

abstract and conclusion. The review of the literature with respect to genomic studies in primary and secondary tumours is welcome. However,

any evidence that doesn’t support the idea that

(i) uniformly size metastatic lesions are associated with genomic similarity

(ii) variably sized metastatic lesions are associated with genomic dissimilarity (with respect to each tumour type) should also be reported.

The readers can then weigh the evidence with respect to the hypothesis noted in #1.

3. However, the most important question remains: how do metastases arise? It is an extraordinarily complex issue, and as the authors rightly point out, there

are many potential confounders that prevent definitive conclusions to be drawn from morphologic data obtained from CT scans with respect to pathogenesis.

Although the study provides evidence associating morphologic patterns to genomic data, the derivation of the pathogenic route of metastatic disease

from genomic data alone, may not, in principle, be possible. For example, inter-tumour metastatic pleomorphism does not necessarily imply early and

pre-clinical seeding events, given the presence of significant genomic instability in many advanced tumours. The authors, I believe, should discuss

these issues in a little more detail.

4. Although the ratio measured by the authors (DPR) seems reasonable, it is interesting that the lowest mean values are ~ 0.5. This is clearly

much larger than the potential lowest value of -1. Does this imply that the linear pathogenic route is ubiquitous? If there is an an element of a parallel

pathogenic route in most tumours, how do we explain - from a biological perspective - both routes arising from the same tumour?

5. While the LPR has the advantage of being simple, are there more complicated measures that could be applied to the patient cohort that may

have added utility? For example, could the longest and shortest axis length of the metastatic tumours be

pooled (with respect to the tissue of origin) and an unsupervised clustering algorithm applied to these data? Investigators could then

simultaneously investigate tumour size and similarity of size. Perhaps different tumour types cluster into specific regions of axis-size space, with varying

sizes of the resultant clusters - we might hypothesise that, for example, metastatic pancreatic tumours cluster into very few clusters, in a particular

region of the space, with negligible intracluster variation. Perhaps the authors, in the discussion, could expand a little on the potential of

analysing metastatic patterning in human cancer.

6. PLOS authors have the option to publish the peer review history of their article (what does this mean?). If published, this will include your full peer review and any attached files.

Reviewer #1: No

Reviewer #2: No

---

## [Author Response · Author response to Decision Letter 0]

25 Jun 2022

Nikolas K. Haass, MD/PhD 23.06.2022

Academic Editor

PLOS ONE

Dear Editor 

Thank you for your letter and for the reviewers’ comments regarding manuscript PONE-D-22-06443, Patterns of Metastases Progression- The Linear Parallel Ratio. We have addressed the reviewers’ comments below, and hope you find this manuscript suitable for publication in PLOS ONE. 

Editorial Comments:

1. Please ensure that your manuscript meets PLOS ONE's style requirements, including those for file naming. The PLOS ONE style templates can be found at https://journals.plos.org/plosone/s/file?id=wjVg/PLOSOne_formatting_sample_main_body.pdf and https://journals.plos.org/plosone/s/file?id=ba62/PLOSOne_formatting_sample_title_authors_affiliations.pdf.

Response: This was corrected as requested.

Response: This was corrected as requested.

Response: This was done.

Response: This was done.

Response: This was done.

5. Please include your tables as part of your main manuscript and remove the individual files. Please note that supplementary tables (should remain/ be uploaded) as separate "supporting information" files.

Response: This was done.

Additional Editor Comments:

I find your manuscript very interesting, as it provides some novel insights into metastasis and the interpretation of CT scans. In accordance with the expert reviewers, there are however, a few concerns that need to be addressed. Rather than repeat those points here, I refer you to the specific remarks (below) for details.

Response: I wish to thank the editor.

---

## [Decision Letter · Decision Letter 1]

5 Aug 2022

PONE-D-22-06443R1

Patterns of Metastases Progression- The Linear Parallel Ratio

PLOS ONE

Dear Dr. Gofrit,

Thank you for submitting your manuscript to PLOS ONE. After careful consideration, we feel that it has merit but does not fully meet PLOS ONE’s publication criteria as it currently stands. Therefore, we invite you to submit a revised version of the manuscript that addresses the points raised during the review process.

This very interesting manuscript provides insights into the metastatic process and can potentially change the way we interpret CT scans. It will certainly be of interest for the community. There are a few minor comments by the expert reviewer, that should be addressed prior to acceptance for publication.

A marked-up copy of your manuscript that highlights changes made to the original version. You should upload this as a separate file labeled 'Revised Manuscript with Track Changes'.An unmarked version of your revised paper without tracked changes. You should upload this as a separate file labeled 'Manuscript'.

We look forward to receiving your revised manuscript.

Kind regards,

Nikolas K. Haass, MD/PhD

Academic Editor

PLOS ONE

Journal Requirements:

Additional Editor Comments (if provided):

This very interesting manuscript provides insights into the metastatic process and can potentially change the way we interpret CT scans. It will certainly be of interest for the community. There are a few minor comments by the expert reviewer, that should be addressed prior to acceptance for publication.

Reviewers' comments:

Reviewer's Responses to Questions

**Comments to the Author**

1. If the authors have adequately addressed your comments raised in a previous round of review and you feel that this manuscript is now acceptable for publication, you may indicate that here to bypass the “Comments to the Author” section, enter your conflict of interest statement in the “Confidential to Editor” section, and submit your "Accept" recommendation.

Reviewer #2: (No Response)

2. Is the manuscript technically sound, and do the data support the conclusions?

Reviewer #2: Partly

3. Has the statistical analysis been performed appropriately and rigorously? 

Reviewer #2: No

4. Have the authors made all data underlying the findings in their manuscript fully available?

Reviewer #2: Yes

5. Is the manuscript presented in an intelligible fashion and written in standard English?

Reviewer #2: Yes

6. Review Comments to the Author

Reviewer #2: I still have an issue with the ideas regarding the hypothesis of this paper.

The authors state that the hypothesis is that the two leading models of metastatic progression can be identified by measuring the diameters of metastases (with the assumption that the leading models of tumour progression are, in fact, true - this latter point needs to made clear). But then how do these data support this hypothesis?

What would the expected distribution of metastatic sizes look like (between and within tumour types) if the leading models of metastatic spread were true? Or false? What if all tumours spread via a mixture of both mechanisms with differences in the variances (with respect to tumour type) of the tumour sizes? And how would that differ if the leading models of metastatic spread were mutually exclusive?

Could the results be simply due to stochastic variation where the two leading models of metastatic progression are irrelevant, or could the results be due to a mixture of both mechanisms of metastatic spread in all tumour types with the main differences between tumour types inherent and history-independent differences in the variances of the tumour sizes?

The problem is somewhat Bayesian. The authors are (sort of) asking Pr(A/B) where A is these data and B is the leading model of tumour spread, but the important quantity is Pr(B/A); that is, the probability of the theory of tumour spread being true given these data. Can the authors provide more rigour in the analysis of their data?

7. PLOS authors have the option to publish the peer review history of their article (what does this mean?). If published, this will include your full peer review and any attached files.

Reviewer #2: No

---

## [Author Response · Author response to Decision Letter 1]

6 Aug 2022

Nikolas K. Haass, MD/PhD 6.08.2022

Academic Editor

PLOS ONE

Dear Editor 

Thank you for your letter and for the thoughtful reviewer comment regarding manuscript PONE-D-22-06443, Patterns of Metastases Progression- The Linear Parallel Ratio. The response to the reviewer comment is below. Minor modifications were made in the manuscript. I hope that after this clarification you will find the manuscript suitable for publication in PLOS ONE. 

Ofer Gofrit

Reviewer comment:

6. Review Comments to the Author

Reviewer #2: I still have an issue with the ideas regarding the hypothesis of this paper.

The authors state that the hypothesis is that the two leading models of metastatic progression can be identified by measuring the diameters of metastases (with the assumption that the leading models of tumour progression are, in fact, true - this latter point needs to made clear). But then how do these data support this hypothesis?

What would the expected distribution of metastatic sizes look like (between and within tumour types) if the leading models of metastatic spread were true? Or false? What if all tumours spread via a mixture of both mechanisms with differences in the variances (with respect to tumour type) of the tumour sizes? And how would that differ if the leading models of metastatic spread were mutually exclusive?

Could the results be simply due to stochastic variation where the two leading models of metastatic progression are irrelevant, or could the results be due to a mixture of both mechanisms of metastatic spread in all tumour types with the main differences between tumour types inherent and history-independent differences in the variances of the tumour sizes?

The problem is somewhat Bayesian. The authors are (sort of) asking Pr(A/B) where A is these data and B is the leading model of tumour spread, but the important quantity is Pr(B/A); that is, the probability of the theory of tumour spread being true given these data. Can the authors provide more rigour in the analysis of their data?

Response:

The metastatic process is extremely complex, and several models attempt to describe it (1st paragraph of the “Discussion”). The leading models are the linear and the parallel models and they are supported by robust molecular data (ref 7-9). As any model in biology, they must be oversimplified, but they do offer a working framework that can be used in research and in the clinic. 

In the current study we do not attempt to prove or disprove these models. We accept them as a basis for the study. The hypothesis of this work (stated in paragraph 5 in the ”Introduction”) is that ”differentiation between the linear and parallel routes of metastatic spread can be done by measurements of metastases diameters on the clinical CT”. We tested this hypothesis on nine tumors that commonly metastasize to the lungs and the results agree with molecular information obtained in complex and expensive experiments. 

The data presented in this manuscript can potentially change our interpretation of CT scans without any additional cost. As mentioned in the “Discussion” the computer code, used in this research, is not necessary in the clinical work since the naked eye is sensitive to this type of information. For example, I saw in the clinic a few days ago a man with lung metastases from gall-bladder cancer (he came to me with urinary retention). His chest CT showed multiple metastases, but it was clear that they could all be fitted into two diameters only. This can be translated to LPR of one with two waves of metastases. As mentioned in the manuscript, this understanding may be relevant in several issues in oncology as in the definition of the oligometastatic state and in management of the primary tumor in a patient with metastatic disease.

---

## [Decision Letter · Decision Letter 2]

8 Sep 2022

Patterns of Metastases Progression- The Linear Parallel Ratio

PONE-D-22-06443R2

Dear Dr. Gofrit,

We’re pleased to inform you that your manuscript has been judged scientifically suitable for publication and will be formally accepted for publication once it meets all outstanding technical requirements.

Kind regards,

Nikolas K. Haass, MD/PhD

Academic Editor

PLOS ONE

Additional Editor Comments (optional):

Reviewers' comments:

Reviewer's Responses to Questions

**Comments to the Author**

1. If the authors have adequately addressed your comments raised in a previous round of review and you feel that this manuscript is now acceptable for publication, you may indicate that here to bypass the “Comments to the Author” section, enter your conflict of interest statement in the “Confidential to Editor” section, and submit your "Accept" recommendation.

Reviewer #2: (No Response)

2. Is the manuscript technically sound, and do the data support the conclusions?

Reviewer #2: Yes

3. Has the statistical analysis been performed appropriately and rigorously? 

Reviewer #2: Yes

4. Have the authors made all data underlying the findings in their manuscript fully available?

Reviewer #2: Yes

5. Is the manuscript presented in an intelligible fashion and written in standard English?

Reviewer #2: Yes

6. Review Comments to the Author

Reviewer #2: (No Response)

7. PLOS authors have the option to publish the peer review history of their article (what does this mean?). If published, this will include your full peer review and any attached files.

Reviewer #2: No

---

## [Editor Report · Acceptance letter]

11 Sep 2022

PONE-D-22-06443R2 

Patterns of Metastases Progression- The Linear Parallel Ratio 

Dear Dr. Gofrit:

I'm pleased to inform you that your manuscript has been deemed suitable for publication in PLOS ONE. Congratulations! Your manuscript is now with our production department. 

Kind regards, 

on behalf of

Prof Nikolas K. Haass 

Academic Editor

PLOS ONE